# The St. Honoré Portal at Amiens Cathedral and Its Reception

## Gili Shalom

Department of Art History, Tel-Aviv University, Tel-Aviv P.O. Box 39040, Israel; gilishal@tauex.tau.ac.il

**Abstract:** This article discusses the depictions of healings on the St. Honoré portal at Amiens Cathedral (post-1240) and the visual strategies by which its viewers were invited to participate in the saint's cult. I contend that the carved figures who gaze or gesture beyond the borders of the tympanum invited the active participation of a broad audience of spectators: male and female, young and old, rich and poor, clerical and lay, and disabled and hale. Moreover, I argue that by referencing both the saint's vita and more contemporary miracle accounts, the sculptures negotiated between the historical past and the Gothic present, allowing the viewers to share in the hope for a miraculous cure.

**Keywords:** Amiens Cathedral; cult of saints; St. Honoré; reception; gothic

Monumental representations of the saints performing healing miracles are quite rare on French Gothic tympana.[1] Mostly based on the saints' vitae, representations of healing either depict the sick interacting with the saint himself or with his shrine. Whereas, in most cases, a single major moment is portrayed to evoke the miracle, the south transept portal of Amiens Cathedral, known as the St. Honoré portal (post-1240), presents no less than three monumentally depicted healings of three different beneficiaries from different social ranks (Figure 1).[2] Sculptured tympana were often perceived as having hermetically closed meanings that might not be discernible to every devotee. They were dubbed "books for the people" or "stone Bibles",[3] requiring their viewers to possess certain interpretive tools or theological teaching in order to fully grasp the visual program. This notion was, however, recently challenged by scholars such as Murray and Jung, who showed that the meaning of Gothic sculptures was much more dynamic than previously argued, and that the portals were viewed from both temporal and spatial aspects (see Jung 2020, pp. 8–58; Murray 2004, pp. 9–12; Rickard 1983, pp. 147–57; Hansen 2015, pp. 105–51. See also Fricke 2015, pp. 93–111). The St. Honoré portal offers an intriguing case study of a visual narrative that is open to its beholders and invites their active participation, particularly via the representations of miraculous healing. While most of the narrative scenes are based on St. Honoré's vita written in the eleventh century (Corblet 1868, vol 3, p. 47; Bolland et al. 1863, May 16, col. 613–15). I suggest that miracle accounts featuring in the contemporary thirteenth-century testimony of Richard of Gerberoy, Bishop of Amiens from 1205 to 1210, also appear to be referenced on the tympanum (Durand 1938, pp. 268–96). Richard, who had himself witnessed the miracles, recounts the healing of a disabled boy and a paralyzed woman following the procession of St. Honoré's relics. I contend that by referencing miracles from the saint's time (6th century) as well as from the medieval present (13th century), the tympanum was able to engage with and intensify the spiritual expectations of the congregants and invite them to actively participate in the sacred cult.

Most scholarly attention to date has been focused on the chronological problems of the construction of the portal and, consequently, on stylistic and iconographic questions (Rivoire 1806; Baron 1900, pp. 61–101; Boinet 1912, pp. 21–48; Katzenellenbogen 1961, pp. 280–90; Erlande-Brandenburg 1977, pp. 253–93; Joubert 2005, pp. 161–70), while the frontal address to the viewer of the healed blind woman on the third register and its connection to contemporary 13th-century events have thus far been neglected. Stephen Murray's *A Gothic Sermon* of 2004 emphasizes the performativity of Gothic sculpture and its interactive character at Amiens Cathedral in particular. Murray explores the relationship

between verbal and visual culture through an analysis of a sermon that was delivered in Amiens during the second half of the thirteenth century. Through this analysis, he convincingly shows that in both sermons and visual apparatus, a similar rhetoric and persuasiveness was used to shape their audiences' behaviors and convey the Catholic dogma to the rural population. Furthermore, Murray argues that the life-sized jamb figures of saints and prophets were not just images from the distant past but served as role models to encourage changes in behavior in the present, actively inviting the beholders' participation (Murray 2004, pp. 9–12). In his more recent book, Murray returns to the locational and historical circumstances of the city during the construction of the cathedral and argues that the transept portals were aimed at helping to distribute both the cult of the Virgin and those of the local saints.[4] Murray also contends that the south transept portal provided principal access for the members of the cathedral chapter as well as a passage for the clergy in procession (Murray 2021, p. 147). Cecilia Gaposchkin too offers a prominent study of how the visual program of Amiens Cathedral was structured around reliquary processions and a connection to the local liturgy. She shows that the sculptured portal not only had a commemorative function but also projected the expectations of the participating audiences, who sought an encounter with the relics of the saints (Gaposchkin 2005, p. 237). Relating to the local history and the iconography of the south transept portal, Hansen shows that the St. Honoré portal created and broadcasted a comprehensive vision of ecclesiastical authority to its audiences through the highlighting of several aspects of the local episcopate and the establishment of an ecclesiastical administration within the diocese (Hansen 2015, pp. 105–51). I will draw on Murray's, Gaposchkin's, and Hansen's investigations of the roles of Gothic audiences in my consideration of the multiple healing scenes and representations of communication on the St. Honoré portal, and show that while these earlier studies have contributed to our understanding of the iconography of the sculptured tympanum, to the question of its function and its varied audiences, they do not discuss the varied means by which such audiences were invited to participate in the sculptured narrative.

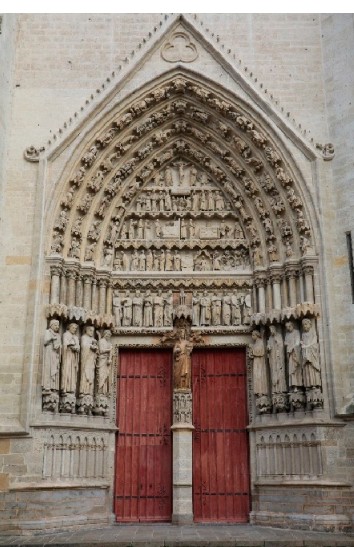

**Figure 1.** Amiens Cathedral, St. Honorius portal, post-1240. (Photo: author).

In this article, I discuss the visual means by which the sculptures of the sick were designed to interact with their contemporary thirteenth-century beholders. I contend that the sculpted figures, whose looks and gestures are directed beyond the borders of the relief, were purposefully introduced to enhance the active participation of those who saw them—especially the disabled beholders of the tympanum. Finally, I argue that the sculptures not only invited their viewers to imitate the represented devotional actions but also helped to merge historical and contemporary time,[5] enabling the Gothic beholders to

share the tympanum figures' gestures and hopes for salvation as well as, most crucially, their miraculous cures.

In order to demonstrate the intended efficacy of the portrayed healing miracles at the St. Honoré portal, I shall first present the textual sources for the iconography of the tympanum. I will then briefly describe the portal as a whole and show how the sculpted figures of the healing miracles were designed to interact with the beholders of the tympanum. I will argue that the sculptures functioned to prepare the disabled for their interaction with the saint's shrine inside the church, fulfilling a teaching purpose. Finally, I shall show that the portal connects the sacred history to the present, merging past and present events. However, before discussing the way in which the historical past and contemporary present are merged in the sculptured tympanum, it is first necessary to become familiar with the saint's vita.

## 1. St. Honoré's Vita

St. Honoré's vita consists mostly of miracle stories from the period of his episcopacy and after his death (Corblet 1868, pp. 38–77; Bolland et al. 1863, May 16, col. 613–15; Hansen 2015, pp. 119–37). He was born in Port-le-Grand in the early sixth century and led a pious childhood. He was proclaimed bishop, and although he refused the honor out of humility, he was miraculously consecrated by a stream of oil that descended upon him directly from heaven. During the first year of his episcopacy, Lupicin, one of his priests, discovered the relics of three martyred saints from the late third or early fourth century. The priest sang a hymn that St. Honoré was able to hear despite being distant from the priest celebrating Mass.[6] The saint witnessed another miracle when, while he himself was performing Mass, the hand of the Savior suddenly appeared and consecrated the host, giving it to St. Honoré. His vita also recounts numerous posthumous miracles effected by the saint, such as the healing of a paralytic boy; the liberation of two prisoners; the healing of a deaf and mute girl; the repair of the damaged skin of a young boy; the healing of a woman who was blind in one eye; the healing of a sick woman from a dreadful disease; the restored sight of a blind woman who had wiped her eyes with St. Honoré's altar cloth; and the healing of a shepherd who was possessed by a demon (Corblet 1868, vol 3, pp. 51–58). St. Honoré's most famous miracle occurred during the translation of his relics from his local parish church to Amiens Cathedral: as the reliquary passed through the nave, the large wooden crucifix on the choir screen suddenly inclined its head towards the saint's remains. Key moments from this vita were chosen to adorn the sculptured tympanum of the south transept portal of the cathedral.

## 2. The St. Honoré Portal

The history of Amiens Cathedral is complex and cannot be studied without considering its political, social, and economic contexts (Kasarska 2011a, pp. 29–46). In brief, in the eleventh and twelfth centuries, the city was characterized by a huge growth in population; improvement in methods of farming and agricultural equipment; the establishment of the textile and dye industries and use of the water mill; and the rise of interest in municipal liberties. These various factors led to the formation of the burgher commune. All members of the commune had to swear an oath of mutual defense and pay an annual fee. The commune's establishment and success led to it becoming responsible for the law and justice in the city, starkly reducing the power of the local seigneur, king, or count, and, as might be expected, hostility and conflict arose (Murray 1996, pp. 20–21). Nevertheless, surviving documents suggest that the clergy often cooperated with the burghers, as evident in the "Respite of Saint Firmin", an annual fee that the commune paid to the bishop to ensure the protection of the burghers.[7] In 1226, this annual fee was reduced by 25%, indicating a close relationship between the burghers and the clergy. The construction of the new Gothic cathedral began within this atmosphere and sense of cooperation, with the joint involvement of the commune, the king, and the clergy (Murray 1996, p. 23).

The construction of the edifice began around 1220, and work on the three portals of the west façade was initiated a few years later.[8] The southernmost portal of the west façade is known as the Virgin's portal and is dated to 1225/1230s; the central Last Judgment portal and the northernmost portal—the St. Firmin portal, are both dated to c. 1230. The St. Honoré portal of the south façade, which faced a major thoroughfare and the canons' houses, is dated to post-1240.[9] It is far more lavishly decorated than the earlier St. Firmin portal. Stylistic differences between the lower and upper parts of the St Honoré portal have led scholars to suggest two phases of construction: one in the mid-1230s, followed by a second one between 1240 and 1269 (Kimpel and Suckale 1973, pp. 217–65).

The narrative of St. Honoré is depicted on five registers upon the tympanum, arranged in chronological order (Figure 2). The lintel below depicts the Twelve Apostles,[10] grouped in pairs, six on each side of the trumeau figure of the Virgin and Child known as the *Vierge Dorée*.[11] The lowest register depicts scenes from Honoré's early episcopacy (Figure 3). On the left half, the seated Honoré is shown receiving the miraculous consecrated oil from above, with six male figures standing in a row in discourse with each other. At the head of the row, two priests are depicted approaching the seated saint. On the right half, Honoré appears again, seated beneath an arched baldachin while Lupicin digs to reveal the tombs of the three saints, as recounted in the vita.

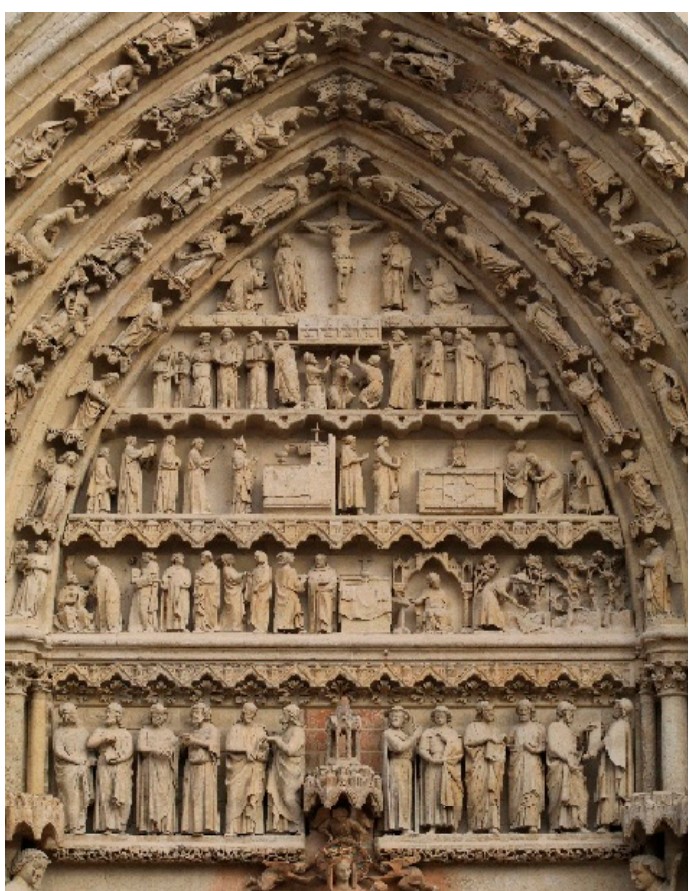

**Figure 2.** Amiens Cathedral, St. Honorius portal, tympanum, post-1240. (Photo: author).

On the left side of the second register, a group of four clerics stand in a row behind St. Honoré, all witnessing the miracle of the hand of God consecrating the host above an altar (Figure 4). On the right side, two male figures in profile are approaching the saint's altar. The garments of the figure on the left suggest that he is a burgher, while the figure on the right can be identified as a cleric holding a book (Leventon 2008, p. 51). The gaze of both men is addressed at the small mitred statue of St. Honoré atop an altar. To the right of the altar stands a young acolyte interacting with a bent figure supporting himself

with a cane (Figure 5). The head of the acolyte is tilted toward the disabled person, as if expressing his empathy. The acolyte is raising the altar cloth to the figure's seemingly closed eyes, likely referring to the healing of the blind woman described in the saint's vita.[12] The body of the woman is depicted in full profile, with her head turned in three-quarter profile, enabling the precise moment of her healing to be accessible to the beholders of the tympanum (Figure 6). This woman can be identified as an inviting figure—a figure that does not belong to the saint's vita, but was introduced into the sculptured event and gazes out towards the beholders, thus communicating with them and consequently merging past and contemporary events—as a historical inviting figure.[13] Behind the woman, an infirm man led by an animal also approaches the altar. Like the burgher and the cleric, he too is depicted in profile, hence enclosing the composition. He may represent the possessed shepherd noted above or another blind man awaiting a miraculous cure (Kasarska 2011b, p. 200).

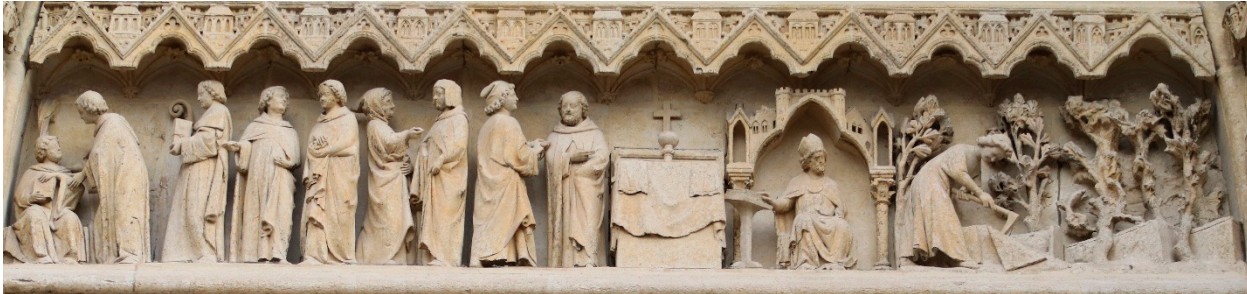

**Figure 3.** Amiens Cathedral, St. Honorius portal, lintel and first register, post-1240. (Photo: author).

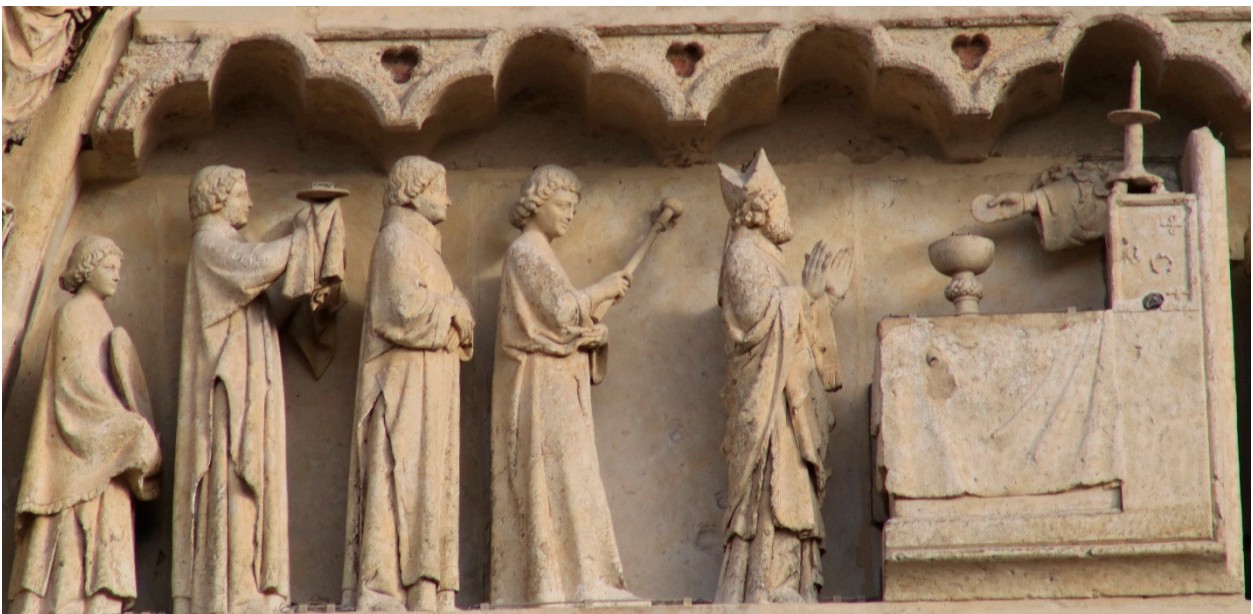

**Figure 4.** Amiens Cathedral, St. Honoré portal, miracle of the hand of God, left side of the second register, post-1240. (Photo: author).

The third register of the tympanum differs from the others in its reading direction and sequence: whereas all the other registers combine several scenes and involve figural movement from both the left and right, the third register's scene begins at the far right and ends at the far left (Figure 7). This register presents the processional rite related to St. Honoré and begins with a boy, whose arm is held by a woman. She is wearing a fillet and barbette, suggesting her high rank (Figure 8) (Davenport 1948, vol 1, p. 167). The boy's head is raised while the woman's is frontal and clearly visible from below despite her recessed position toward the background.[14] A cluster of six figures appears to the left.

The three figures in front, depicted in profile and walking behind St. Honoré's châsse, partially obscure two men and a woman standing in the background, looking in various directions. The reliquary itself is shown borne on a litter supported by two men.[15] Other than the woman with the boy, none of these processional figures look or gesture at the beholders. Crouched below the reliquary, three other figures are portrayed hoping for a miraculous cure (Figure 9). One man is seen from the back, touching the châsse with the fingers of his left hand, while his right hand holds a (now broken) cane. His disability is suggested by his twisted legs, which seem unable to support his upper body. The lower part of the central figure is frontal, and his legs are also depicted as bent while his torso turns inwards, supported by a cane to emphasize his disability. This figure's other hand is reaching out to touch the châsse from below, with the entire palm of his hand. The third crouching figure to the left is a bearded man depicted kneeling in profile and praying while he looks up at the reliquary. At the head of the procession, a group of three acolytes and three choir boys are holding liturgical objects including prayer books, an arm reliquary,[16] and a candle. Although none of the figures look beyond the register's surface, the arm reliquary is frontally presented to the beholders by the acolyte. This complex arrangement of processioning figures offers a complete image of the type of audiences that participated in St. Honoré's cult—young and old, injured and healthy, male and female, and lay and clerical. The top register of the tympanum depicts the Crucifixion, while concomitantly evoking the crucifix that bowed to St. Honoré during the procession, as recounted in the saint's vita. Mary and John the Evangelist are portrayed on either side of the Cross together with two kneeling angels (Figure 10).

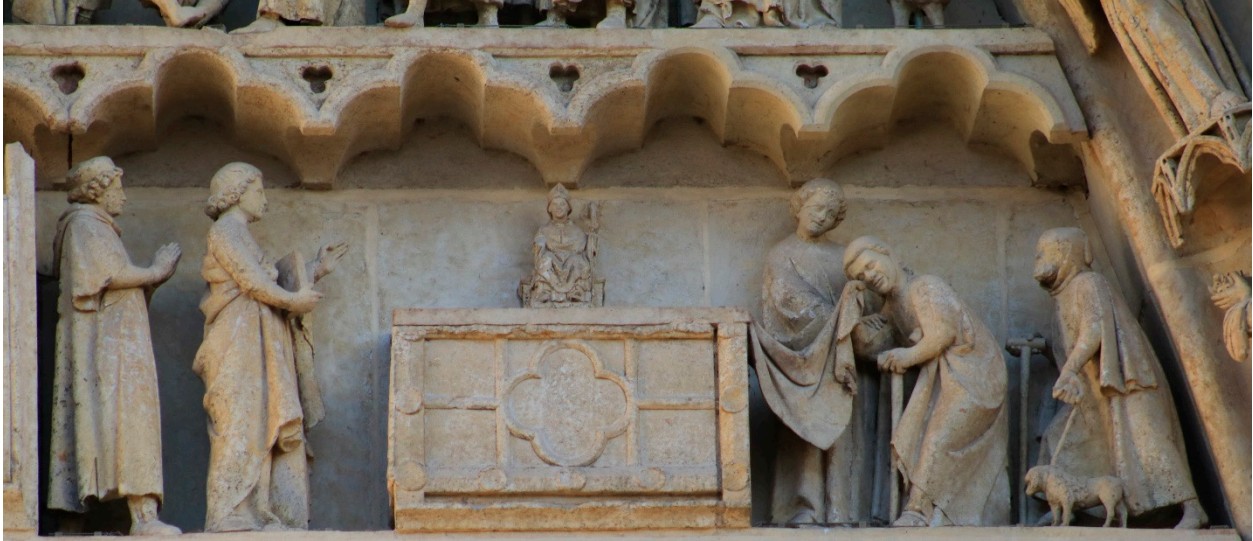

**Figure 5.** Amiens Cathedral, St. Honoré portal, healing of a blind woman, right side of the second register, post-1240. (Photo: author).

The iconographic program of the St. Honoré portal showcases key moments from the saint's vita, as one might expect. Throughout the scenes, however, it is principally the beneficiaries of the healing miracles who engage with the viewers of the tympanum through their direct looks or gestures that invite the viewers' participation in the narrative: the mother with her son, the injured men beneath the châsse, and the blind woman who wipes her eyes with the altar cloth. All these figures serve to link the external viewers of the tympanum with the power of the saint in the imagery. First, the raised head of the boy suggests he is responding both to the saint's holy power and to his mother behind him, in a composition that creates a triangle between the beholder of the tympanum and the two protagonists, involving the beholders, in St. Honoré's miracle. Second, the disabled men under the châsse, presenting a variety of bodily positions and angles while approaching and touching the shrine, are directed to the beholders, indicating the range of ways by

which the saint's aid could be petitioned—ways that the viewers may emulate. Third, the historical inviting figure, in this case the blind woman, is depicted at the moment of her cure as the altar cloth is miraculously elevated in the air and delicately touches her eye, is turning her head to look expressly beyond the tympanum surface with her healed gaze, making contact with the sculpture's viewers. Such a range of gestures and glances from different angles and from a variety of protagonists opens up the narrative to a wide variety of audiences and invites the spectators to seek and experience the saint's efficacy.

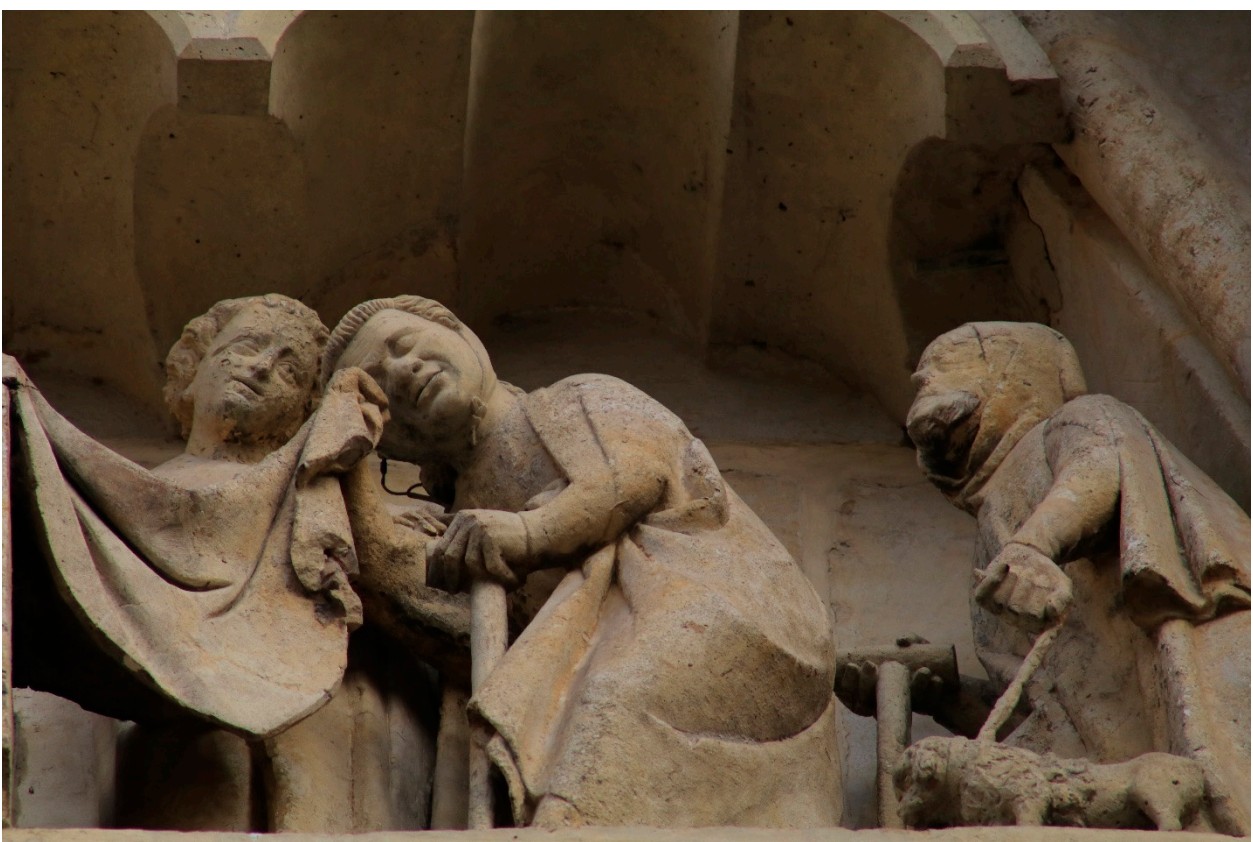

**Figure 6.** Amiens Cathedral, St. Honoré portal, healing of a blind woman (detail), right side of the second register, post-1240. (Photo: author).

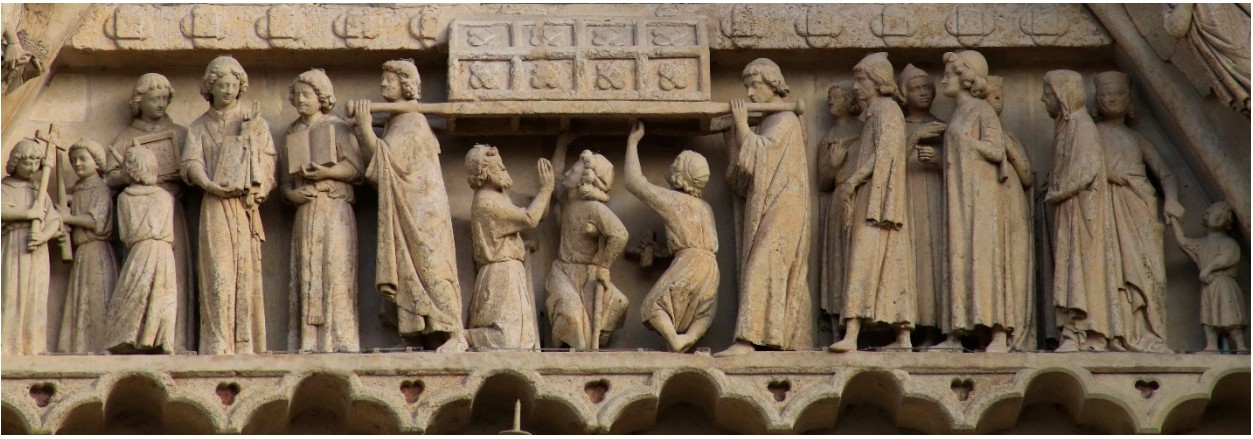

**Figure 7.** Amiens Cathedral, St. Honoré portal, processional rite, third register, post-1240. (Photo: author).

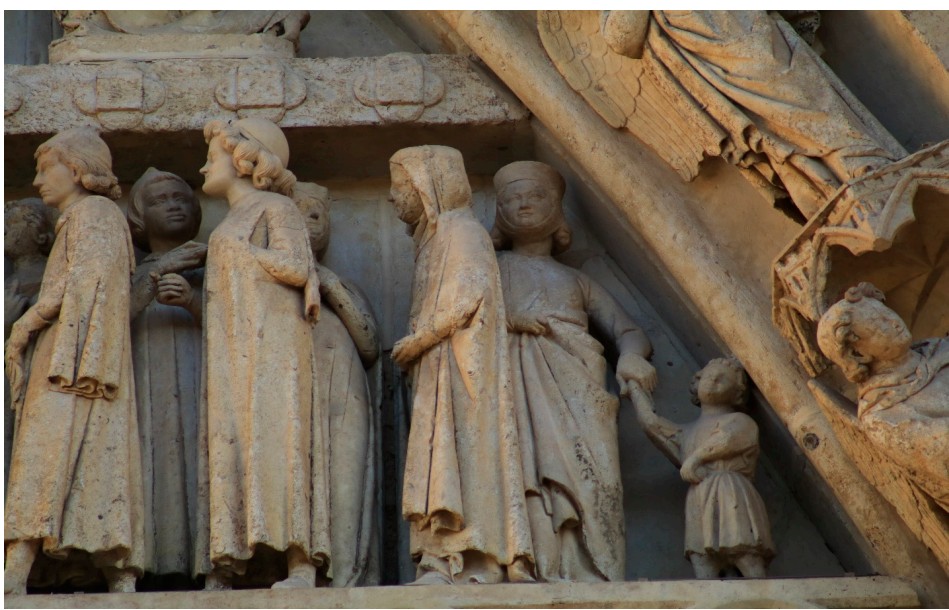

**Figure 8.** Amiens Cathedral, St. Honoré portal, woman with a boy, third register, post-1240. (Photo: author).

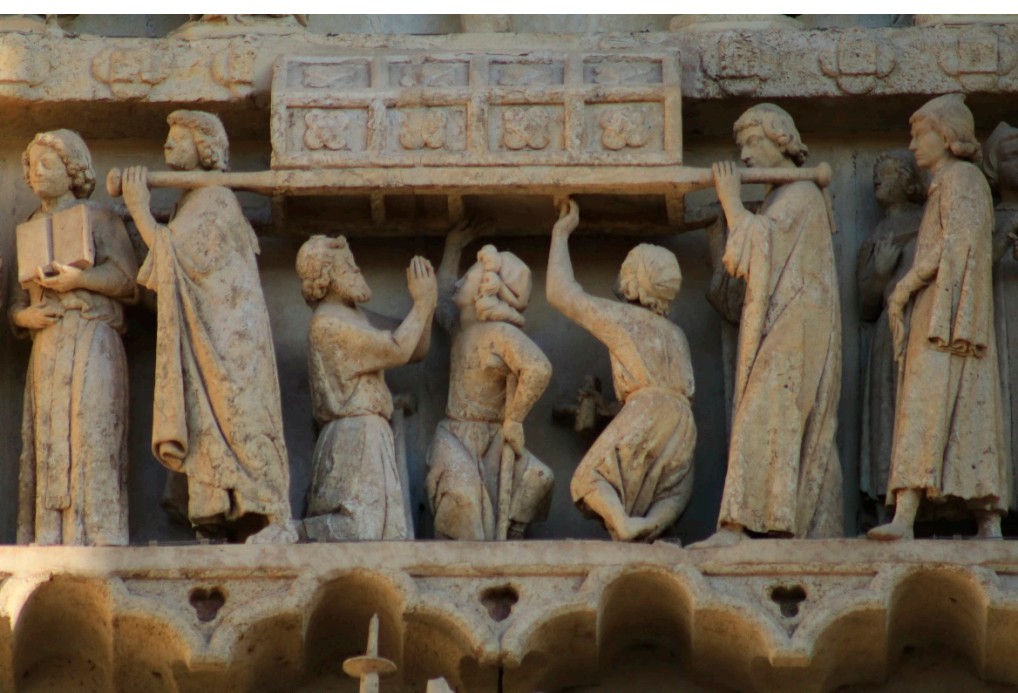

**Figure 9.** Amiens Cathedral, St. Honoré portal, three disabled men with St. Honoré's châsse, third register, post-1240. (Photo: author).

In other depictions of healing miracles elsewhere, the main figures are portrayed more passively awaiting a miracle. For example, in the confessors' portal at Chartres Cathedral (c. 1215–1220), five devotees wait below St. Martin's shrine to receive the oil and water that flow from his tomb and that can heal them (Figure 11) (Katzenellenbogen 1959, pp. 79–90, esp. 81). Although the heads of two of the figures seem to be angled beyond the tympanum surface, one of the figures is actually represented in the process of applying the holy oil to her head while the other figure is simply sleeping below the shrine, waiting for the miracle.[17] Here, in contrast to Amiens, the figures stand by, patiently waiting for a sign from above; there is no depiction of the healed, but only of those in need of healing.[18]

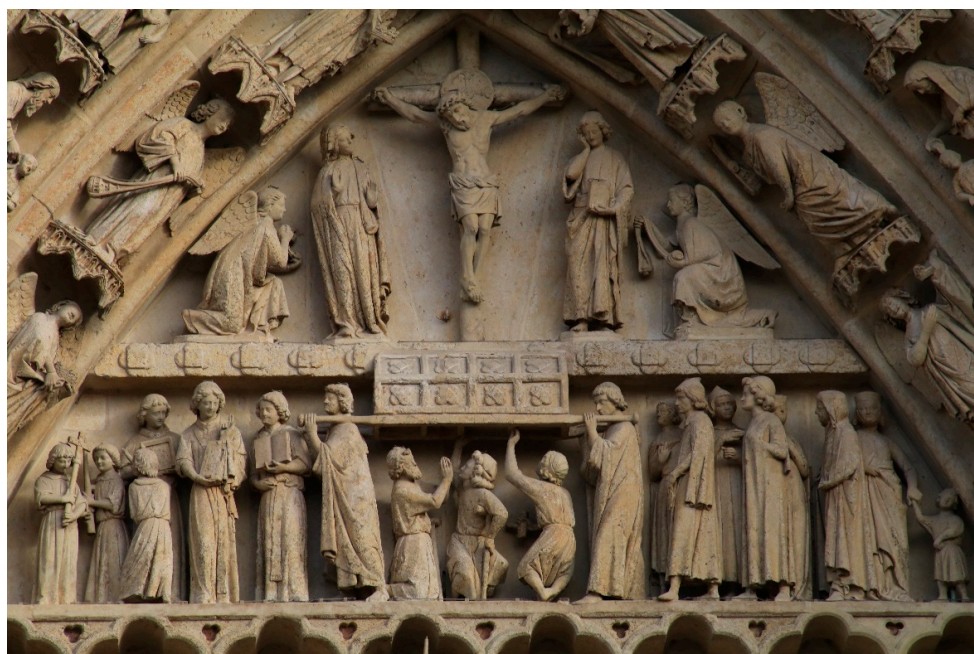

**Figure 10.** Amiens Cathedral, St. Honoré portal, Crucifixion, fourth register, post-1240. (Photo: author).

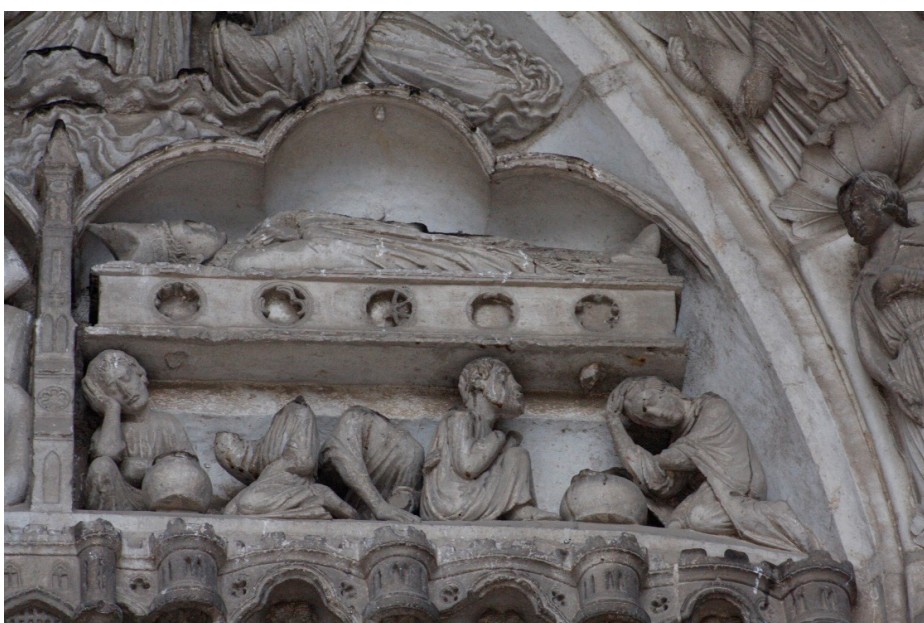

**Figure 11.** Chartres Cathedral, confessors' portal, eastern portal of the south façade, St. Martin's healing miracle, c. 1215–1220. (Photo: author).

In the portal of the Saints at Reims Cathedral (c. 1220), the only healing miracle is that of St. Remi healing the girl from Toulouse.[19] The fragmented narrative is spread over three registers and comprises the girl's exorcism, St. Benedict's attempt to heal her, and her resurrection by St. Remi. However, in none of the scenes does the girl face the beholder or gesture bodily in a way that opens up the narrative to the audience (Figure 12).[20]

These comparisons reveal the very different and striking nature of the scenes in Amiens, in which the depicted sick figures actively engage with the beholders via eye contact and gesture. The idea of the Amiens figures as role models is particularly expressed in the figure of the injured man beneath the châsse whose back is facing the viewers. His position and upward gaze exactly mirror that of the viewers, looking up at the tympanum.

As such, the sculptures addressed those beholders who, like the sculptured devotees, were seeking miracles and were active participants in St. Honoré's cult.

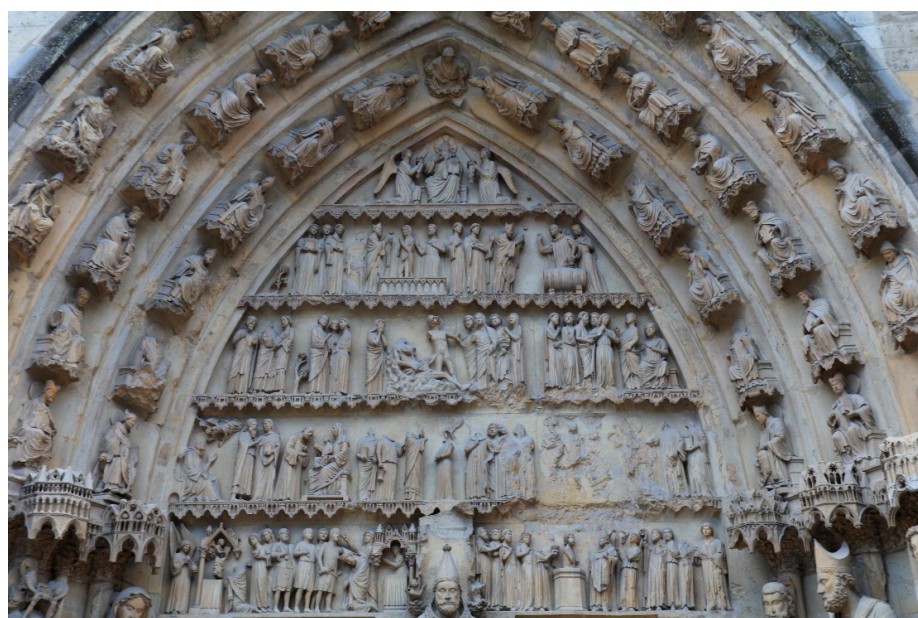

**Figure 12.** Reims Cathedral, portal of the saints, c. 1220. (Photo: author).

### 3. Miracles and Reception of the St. Honoré Portal

In her study of the reception of the Amiens portals, Gaposchkin explores how the visual program of the two tympana was structured around reliquary processions and their connection to the local liturgy. She anchors her analysis in the liturgical rites of St. Firmin and St. Honoré, which took place in Amiens several times a year. During the processions for these rites, the reliquary châsses of the saints were carried beyond the cathedral wall and through the cloister or out into the city and the surrounding countryside. Processions were held both on feast days (i.e., "ordinary" processions) and as fund-raising efforts to support construction (i.e., "extraordinary" processions or "relic-quests"). St. Honoré relic-quests were performed in 1060 and in 1240 and also in times of plague, flood, or drought.[21] Gaposchkin suggests that these processions entered and exited through the same doors on which the sculptures of the saints were represented, and the procession often included opportunities for lay involvement in the ecclesiastical rituals, as can be learned from the thirteenth-century testimony of Bishop Richard of Gerberoy:

> On the great day of the feast of that confessor [St. Honoré], . . . we, a great crowd composed equally of clergy and laymen, were accustomed to carry the body of that saintly confessor around our cloister, after we had returned . . . the reliquary of the most blessed confessor was placed upon the altar in the middle of the church [and] was protected by the devoted services of the assisting priests [and] the people (*plebs*) . . . The aforementioned [crippled] boy approach[ed] the reliquary of that blessed confessor, and look[ed] at those [people] who were looking at [the reliquary], as if he was about to receive something from them; he sensed within himself a hidden miracle of divine power, which was apparent to all others in its workings. Indeed, his mother being there and insisting on prayers, the boy began to proclaim that it seems to him to come from on high . . . The mother . . . respond[ed] to him: "Have faith, my son, and get up, touch the reliquary of the blessed confessor. I believe that you will be saved by the one who gives aid." At these words the boy got up, and his feet and his arms were equally restored in their strength; we . . . resounded in praise of the Lord and sounded the bells of the church.[22]

While Gaposchkin refers to the bishop's account to show that the reliquary was carried by both clerics and laymen, as is represented on the tympanum, she did not notice that the story of the crippled boy and his mother is included on the third register of the tympanum, to the far right. On the tympanum, the boy's raised head suggests that he is sensing the power of the saint, while his mother in a barbette faces frontally and clasps her child's newly healed arm. I believe that the mother is represented as addressing the beholders directly in order to extend the invitation to participate in St. Honoré's cult specifically to wealthy parents seeking cures for their children. Moreover, the fact that the child is not depicted as crippled implies both that the miraculous cure was successful and that every child in need can participate in the processions and pay reverence to the saint. This detail thus augments the message of the processional frieze and of Bishop Richard's account that all manner of devotees—men and women, young and old, wealthy and poor—can interact with and benefit from St. Honoré's shrine.

The bishop's account also mentions a paralyzed woman who responded to the miraculous healing of the boy:

> The news of the miraculous healing spread fast in the hospital located near the cathedral ... There, an unfortunate woman paralyzed in all her body except her hands was condemned to stay in her bed; when she heard the bells ring to celebrate the miraculous cure (of the boy), she felt inspired with a sudden confidence, and was transported to the cathedral. The woman encircled with her hands the protective châsse and found herself miraculously healed. In the presence of the bishop she blessed God for his act, and after receiving some donations from the faithful, she went back to the hospital in order to share her joy with all of whom have sympathized with her sufferings.[23]

This thirteenth-century testimony of Richard of Gerberoy sheds additional light on the question of how the tympanum was received by the local audience. First, it reveals that the miracle of healing the paralyzed boy was well known and aroused great interest at the time, which may explain why the planners of the iconographic program chose to make reference to it in a tympanum that supposedly presents key moments from the eleventh-century vita of St. Honoré. Second, although Bishop Richard's text does not reveal through which portal the paralyzed woman had entered in order to interact with St. Honoré's shrine, we do know that the processions passed through the south transept portal at Amiens. This makes it likely that the woman also entered the cathedral via the St. Honoré portal, albeit before the new portal program was added, which, according to Murray's latest monograph, occurred between 1230 and 1240 (Murray 2021). It is possible to argue that the tympanum had a priming role for those in need who came seeking a miracle, as it not only encouraged them to actively participate in the narrative but also demonstrated how to approach the shrine and physically engage with it. Third, both the paralyzed woman and the sculptured figures beneath the reliquary are healed through the act of touch. The accessibility of the shrine to a wide range of disabled devotees implies that healing is possible for every individual, echoing the inclusive message. I believe that such a tactile emphasis stresses the saint's approachability and interactivity,[24] as is also emphasized by the joint lay and ecclesiastical processional rites and the donations.[25] Thus, the beholder could relate to the sculptures in two ways: on the one hand, through sight, which was understood in the Middle Ages as either an active or passive action. Known as the extramission and the intromission theories, respectively, these two optical theories were a subject of debate in the High Middle Ages. In the intromission theory, rays leaving an object enter the eye, imprinting its form on the merely receptive eye. Thus, the eye is a passive receptor of light, and the beholder's role is as the passive recipient of external realities. In the extramission theory, visual rays are emitted from the eye and are enhanced by the presence of light. The rays then meet the object and are shaped by it, only then returning to the eye. Thus, sight is an active process, involving a direct and mutual physical relation between subject and object. During the thirteenth century, extramission was the dominant optical theory, and it claimed a direct connection between subject and object, a performative and interactive spectatorship in

which the viewer was an active agent in the process of seeing and knowing. Within this interplay between sight and touch lie the subjects of textures and materiality, which cause the beholders not only to see but also to touch in their mind's eyes—both metaphorically and literally.[26]

Finally, the combination of the healing scene of the young boy, which references cures closer to the time of the portal's construction, together with the portrayals of the earlier miracles that had taken place during St. Honoré's life and his translation, created a bridge between hagiographic history and the present time of the beholders. Because both the sculpted participants in the relief and the actual beholders could walk in the same processions, share the same gestures, and reach out to touch the same reliquary, the hope for miracles was shared between the past and present at this holy site. As is clear from the thirteenth-century text, the diverse audiences of the relic processions were invited to enter and actively participate in the events with the saint, just as beholders of the tympanum were invited, by the figures catching and holding their gazes, to enter into the historical moment and relive it in their present time.

## 4. Conclusions

In his recent study of the narthex portal of Vézelay, Rudolph has offered a new understanding of the iconography of the tympanum, and argued that "the intended audience of the complex program could not have understood its meaning to any appreciable degree except through a guide, who both mediated it and allowed the complexity, in the process facilitating institutional control in the non-elite participation in elite spirituality" (Rudolph 2021, pp. 601–61). In contrast to the Romanesque portal, the St. Honoré tympanum offers an illuminating case study of the power of images to engage with their varied audiences and show them how to relate to the cult of the local saint without being dependent on a guide. Regardless of the viewers' gender, status in life, membership in the sacred or secular hierarchy, or physical ability, the sculptures address them all equally through looks and gestures, creating a sense of community among those who enter the cathedral via the south transept portal (Van Gennep 1909, p. 156). Moreover, it can be argued that the guide was replaced here by the inviting figures, which have played a key role in our understanding of the reception of medieval artworks by their contemporary beholders: shifting their identification from *passive* recipients, whose ability to decipher the scenes was limited and dependent on the interpretations of literate and educated commentators, to *active* participants, whose own personal, experiential understanding of the scenes played a pivotal role in the construction of meaning. Implanted in the scene for a communicative purpose, such inviting figures offered the beholders personal and intimate access to the sacred history. They thereby did not merely facilitate the understanding of the narratives but, more crucially, through their direct gazes and vivid gestures, they left the open narratives and unsolved plots to be completed by the beholders according to their own experiences. This visual stratagem resulted in the active involvement of the audience and turned the sculptured tympanum into an interactive environment. Consequently, it seems reasonable to reject Marcel Proust's understanding of Amiens's sculptures as "arcane hieroglyphs" and to accept instead John Ruskin's romantic reading that the St. Honoré portal is a kind of open book, accessible to all.[27] As such, these Gothic portals are not merely "Posters in stone" but, perhaps, "Living posters in stone" (Sauerländer 1992, pp. 17–44).

**Funding:** This research received no external funding.

**Institutional Review Board Statement:** Not applicable.

**Informed Consent Statement:** Not applicable.

**Data Availability Statement:** No new data were created or analyzed in this study. Data sharing is not applicable to this article.

**Conflicts of Interest:** The author declares no conflict of interest.

## Notes

1   The only two other contemporary examples are the Confessor's Portal at Chartres (c. 1215–1220) and the Portal of the Saints at Reims (c. 1220), which is briefly discussed below.

2   There has been scholarly debate on the dating of the sculptured tympana. Murray argues that the construction of the portal began in the 1230s and that its sculpture work is dated to post-1240, except for the jamb figures (Murray 1996, pp. 118–20). Kimpel and Suckale suggest a pre-1236 dating for the beginning of construction (Kimpel and Suckale 1985, pp. 11–64). Murray dismissed this idea. Finally, Gaposchkin dates the sculptures to 1260, which is the date broadly accepted today for the end of the construction of the portal and the insertion of the trumeau figures (Gaposchkin 2005, pp. 217–42). I find Murray's dating the most persuasive.

3   See Mâle (1913, pp. 390–96, esp. 398). Mâle noted: "The medieval cathedral takes the place of books" (ibid., 398). The idea of the sculptural program of the cathedral as a "Stone Bible" was first introduced by John Ruskin in connection with Amiens Cathedral (Ruskin 1908, pp. 5–187).

4   See Murray (2021, pp. 146–55). See also the website "Life of a Cathedral: Notre-Dame of Amiens" Available online: https://mcid.mcah.columbia.edu/art-atlas/mapping-gothic. (accessed on: 10 April 2024)

5   In the most immediate way, the historical time is the past time where the action took place. The contemporary time is the time of the thirteenth-century beholder. As in the case of the Baptism of Clovis at Reims Cathedral, the beholders are not only invited to participate in the past event and to revive it, but also offered them the possibility of reliving it in the present procession, through movement in space and time, as part of the ceremony. See Shalom (2017, pp. 96–113). See also Jaritz and Moreno-Rianō (2003).

6   The relics of the three saints were later offered as a gift to the cathedral (Beauvillé 1867, p. 11).

7   A representation of this fee can be seen in the middle register of the St. Firmin portal, in which one of the boys holds a coin.

8   The 1220 dating for the beginning of the construction is based on two written sources: the inscription around the edge of the bronze tomb of the founding bishop, Evrard de Fouilloy, and the inscription in the center of the labyrinth on the nave pavement. However, as Murray shows, these sources do not clarify when the work started and how it progressed. Today, most scholars agree that construction began in the nave and continued towards the choir. In 1206, the church received the head relic of St. John the Baptist from Wallon of Sarton, who was the canon of Picquiny. However, it seems that it was the 1218 fire that led to the construction of the new edifice. See (Murray 1996, pp. 26–27; Kimpel and Suckale 1973, pp. 217–65).

9   See also note 2. It is generally accepted to distinguish the "hands" of three masters who presided successively over the construction work: Robert de Luzarches (d. 1223), whose work included the dado wall of the south transept portal, the nave, and the west façades; Thomas de Cormont (d. 1228), who introduced his own forms in the tracery of the choir aisle window, the dado in the radiating chapels, and the transverse arches and capital in the ambulatory and upper nave; and Renaud de Cormont (n.d.), Thomas's son, who constructed the upper transept and the upper choir. The names of the three masters along with the name of the founding bishop, Evrard de Fouilloy, appear in the labyrinth dated to 1288. For the question of the portal's location, see Murray (1996, p. 118). For studies of Amiens's labyrinth, see (Soyez 1984; Bord 1976).

10   There is disagreement regarding the identification of this scene: Katzenellenbogen suggests that the people represent types for the function of the bishop, similar to the bottom register of the St. Firmin portal (Katzenellenbogen 1961, p. 281); Murray recognizes them as the Twelve Apostles (Murray 1996, p. 119); while Sandron (2004) argues that the scene depicts the separation of the Twelve Apostles, which might explain their shoeless state (p. 139).

11   Originally, the trumeau figure depicted St. Honoré (Kasarska 2011b, p. 200). The famous *Vierge Dorée* dates to 1288. The one in situ is a copy of the original and is located inside the cathedral. See Sandron (2004, p. 138).

12   It is hard to tell if the figure is indeed male or female. It has no breasts and the hair is tied under a coif, suggesting its masculinity. However, none of the other female figures too in the portal have breasts (for example, the mother holding of the boy by his arm), and the textual source clearly states that it was a woman whose sight was miraculously restored. Therefore, it seems plausible that the tympanum follows the text and depicts a woman.

13   On the different types of inviting figures see (Shalom 2017, pp. 96–113; Shalom 2019, pp. 89–113).

14   Additionally, it should be remembered that the sculptures were once painted. On the polychromy of the portal, see the final four essays in Verret and Steyaert (2002, pp. 207–58).

15   Little is known about this reliquary, which was lost during the French Revolution (Durand 1901, p. 2)

16   The arm reliquary also connects the tympanum with that of St. Firmin. See Gaposchkin (2005, p. 240). For the role of arm reliquaries, see (Boehm 1997, pp. 8–19; Hahn 1997, pp. 20–31).

17   The broken, fragmented sculptures just above this figure perhaps depicted the flow of the holy oil descending from above. However, the current state of the sculpture makes it impossible to confirm such a reading.

18   On the question of the audience of the portals in the south porches, see Hollengreen (2004, pp. 81–108).

19   On the story and its textual sources, see Hinkle (1965, p. 48).

20   In contrast, the two scenes, in Reims on the lintel, the Baptism of Clovis and the Martyrdom of St. Nicaise, invite the active participation of their beholders. See Shalom (2017, pp. 96–113).

21    St. Firmin's principal feasts commemorated the translation of his relic on January 13; his martyrdom on September 25; his octave on October 2; his arrival at Amiens on October 10, and the deposition of his relics in a new reliquary on October 16. St. Honoré's principal feast and octave were on May 16. See Gaposchkin (2005, pp. 222–23).

22    Bolland et al. (1863, May 16, col. 613–15): *In die etenim magno festivitatis praedicti Confessoris, qua nos, cum multa cleri pariter; populi frequentia, corpus sanctissimi Confessoris circa claustrum nostrum deferre consuevimus, redeuntibus nobis ad ecclesiam, cum jam clerus in chorum se recepisset missarum celebraturus solennia; theca beatissimi Confessoris in medio ecclesiae super tabulam posita, sacerdotum assistentium devotis servaretur obsequiis; plebs, ibidem in honore Domini; beati Confessoris collecta, orationes suas diutius funderet; vota solveret repromissa; puer praedictus ad thecam beati praedicti confessoris accedens; intuens in eos qui eam observabant, tamquam aliquid accepturus ab eis; sensit intra se divinae SIC? virtutis occultum miraculum, quod foris omnibus manifestum est in opere. Praesente enim matre sua; orationibus insistente, clamare coepit puer, quod videbatur ei si sursum trahi: sed parum intelligens, nondum cognoscebat quae sibi virtus divina praeparasset. Cui mater in Domino confortatt SIC?, respondit: Confide, fili, surge, apprehende thecam beati Confessoris. Credo enim quod ipso opitulate salvaberis. Ad quae verba surrexit puer; pedum pariter ac tibiarum recuperata virtute, in laudem Dei pariter conclamantes; ecclesiae pulsantes classicu; nos qui in choro eramus prius stupore.* Translation from Gaposchkin (2005, pp. 235–36, n. 70).

23    "La nouvelle de cette guérison subite parvint bien vite à l'hôpital voisin de la cathédrale où, sous la direction de frères hospitaliers, la charité des fidèles nourrissait des pauvres et des infirmes. Là, une malheureuse femme perclue? CHECK de tous ses membres, à l'exception des mains, était condamnée à rester toujours au lit; en apprenant le miracle qui célébrait la voix joyeuse des cloches, elle se sent inspirée d'une subite confiance, se fait transporter à la cathédrale, entoure de ses bras la châsse protectrice et se trouve soudain guérie; elle bénit Dieu de son bienfait, en présence de l'évêque, et, après avoir reçu quelques libéralités des fidèles, elle retourna à l'hospice pour faire partager sa joie à tous ceux qui jadis avaient compati à ses souffrances." The bishop's testimony postdates the vita and was added in the vernacular French. My translation is based on Corblet (1868, vol 3, p. 55). On the structure of the medieval city of Amiens and the location of the hospital, see Bayard (1999, pp. 199–214).

24    On the question of the senses in Gothic art, see the studies in Palazzo (2016, pp. 547–647). On Tactility see Jung (2010, pp. 203–40)

25    I do not wish to suggest that seeing is less important than touching but, rather, that the sculptures' emphasis on touch was an essential aspect of curative miracles. For seminal studies on theories of sight and its importance in medieval art, see (Lindberg 1976, pp. 42–52; Caviness 2001, 2000; Elsner 1995; Nolan 1977; Kessler 2000; Nelson 2000 and Biernoff 2002).

26    The interplay between sight and touch has been widely explored in the last five decades, but this topic is beyond the scope of the present article. See the seminal study by Nelson, Hahn, and Camille in Nelson (2000).

27    Proust (1986) states: "Il n'y a pas de Logos, Il n'y a que les hiéroglyphes" (préface, pp. 44–45). See also (Deleuze 1993, p. 195; Ruskin 1908, pp. 249–340).

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
