# Peer review of "The St. Honoré Portal at Amiens Cathedral and Its Reception"

_religions, doi:10.3390/rel15050536_

Round 1

Reviewer 1 Report

Comments and Suggestions for Authors

Overview

Focusing on the depictions of healings on the St. Honorius portal at Amiens Cathedral, the author suggests exploring the visual strategies by which its viewers are invited to participate in the saint’s cult. While the study thoroughly examines primary sources and visual imagery, it could benefit from a more in-depth exploration of core aspects such as vision, tactility, and the perception of time. I would recommend this article for publication after revision.

Here are my suggestions:

There is an extensive scholarship on the reception of sculpted religious portals. I advise the author to read carefully, digested, and intellectually integrated into the essay's argument. See, for example, Conrad Rudolph, “Macro/Microcosm at Vézelay: The Narthex Portal and Non-Elite Participation in Elite Spirituality,” Speculum 96.3 (2021): 601–61.

Jung, Jacqueline E. Eloquent bodies: movement, expression, and the human figure in Gothic sculpture. Yale University Press, 2020.

At the end of p. 2: The paragraph describing the work of Murray and Gaposhckin is ended with the sentence, “I will draw on Murray’s and Gaposhckin’s investigations of the roles of Gothic audiences in my consideration of the multiple healing,” which might seem better if s/he explains in what way and for what end.

On p. 3, the author declares, “I argue that the sculptures not only invited their viewers to imitate the represented devotional actions but also helped to merge historical and contemporary time….” the central issue of "time" should be thoroughly addressed. The author should explore the perception of time and its utilization in the interaction between viewers and the sculpted narrative.

A good starting point could be: Jaritz, Gerhard, and Gerson Moreno-Riaño, eds. Time and Eternity: The Medieval Discourse. Brepols Publishers, 2003.

The sub-chapter, St. Honorius’s Vita, in p. 3 it appears somewhat disconnected from its current placement. The author might consider relocating it to page 4, possibly before the section on “The narrative of Honorius is depicted on five registers upon the tympanum.” Alternatively, they could find another suitable solution to integrate it more cohesively into the overall flow.

In note 37, the author refers to theories of sight and tactility. It is advisable to incorporate a comprehensive discussion of these issues throughout sections examining the visual narrative's reception. This is a crucial issue that deserves elaboration an

Author Response

I wish to thank you for the insightful and useful comments on my article “The St. Honoré Portal at Amiens Cathedral and Its Reception”, which have helped me to further develop and present my arguments in greater depth. I have read all the comments carefully and introduced the changes/additions where needed.

  1. The essay by Conrad Rudolph is now inserted to the conclusion.
  2. All the studies by Jung, Murray, and Hansen have now been addressed in the article (pp.2 & 3); and I have also explained in what way and to what end I draw upon Murray’s and Gaposchkin’s studies (pp. 3-4).
  3. Regarding the subject of time: I believe that a discussion of this theme would draw the readers’ attention away from the theme of reception. However, I now explain my meaning of the term “contemporary time” as follows: “in the most immediate way, the ‘historical time’ is the past time in which the action took place; while ‘contemporary time’ is the time of the thirteenth-century beholder. As in the case of the Baptism of Clovis at Reims Cathedral, the viewers are not only invited to participate in the past event and to revive it, but are also offered the possibility of reliving it in the present procession, through movement in space and time, as part of the ceremony. I have now added the following two references for a further understanding: See Shalom, “Reliving the Past in the Present,” 96–113. See also Jaritz, Gerhard, and Gerson Moreno-Riaño, eds. Time and Eternity: The Medieval Discourse. See note 13 on p. 4.
  4. The sub-chapter on St. Honorius’s vita has now been integrated more smoothly into the overall flow (pp.4-5).
  5. I have addressed the subject of sight and touch in the text (pp. 15-16) and added additional references in footnote 45.

Reviewer 2 Report

Comments and Suggestions for Authors

Summary

The manuscript under review focuses on the tympanum of the St. Honoré Portal on the south transept façade of Amiens Cathedral. The author argues that, far from being geared exclusively toward an elite clerical audience, a diverse range of viewers could appreciate the thaumaturgical miracles depicted on the portal. While previous scholarship has already established that the saint’s vita inspired many scenes, the author of the present study proposes that an account contemporary with the portal’s construction may have inspired a detail on the right side of the third register. The manuscript is clear, relevant to the field of Gothic art and architecture, and structured nicely.

General Comments

While many of the key studies of Amiens Cathedral portal sculpture appear in the notes, there are additional references I would expect to see in an article on this topic, including Stephen Murray’s most recent book about Amiens Cathedral, Notre-Dame of Amiens: Life of the Gothic Cathedral (New York: Columbia University Press, 2021), esp. pp. 154-155; the website “Life of a Cathedral: Notre-Dame of Amiens,” <learn.columbia.edu/amiens>; the section on the St. Honoré portal in Lindsey Hansen’s dissertation, “The Bishop Performed: Sculpture, Liturgy, Episcopal Identity in Thirteenth-Century France,” PhD diss. (Indiana University, 2016), pp. 105-151; and, on interactivity in Gothic monumental sculpture, Jacqueline Jung’s Eloquent Bodies (New Haven: Yale University Press, 2020). Above all, this recent scholarship on Gothic sculpture has already challenged the notion that “sculptured tympana are often perceived as having hermetically closed meanings”(1), a claim that comes across at this point as something of a straw man. 

The idea that the Amiens sculptural program invites viewers in is a compelling one, but I would argue that it is not necessarily limited to the south transept portal. I recommend considering the extent to which this case study extends an existing practice evident already in the sculptural program of the western frontispiece. To my mind, the gesture used to expel Adam and Eve from Eden, for example, at the base of the trumeau of the south portal engages the viewer and harnesses the architecture of the macro in service of the micro, eliding the biblical past with the viewer's present. 

Specific Comments

“Whereas most scholarly attention has been focused on the chronological problems of the construction of the portal and, consequently, on stylistic and iconographic questions, the communicative strategies of the sculptural program of St. Honorius’ portal has thus far been neglected”(2). Consider rephrasing the opening sentence of this paragraph, as the rest of the paragraph seems to contradict it. 

"This fact makes it likely that the woman also entered to the cathedral via the St. Honorius portal, albeit before the new portal program was added"(13). Murray's most recent assessment of the chronology of the south transept portal in his 2021 monograph should be incorporated into this argument to make it more specific.

Since there is also a south portal on the western frontispiece of Amiens Cathedral, I recommend changing “south portal” to “south transept portal” throughout the manuscript.

Since there is another St. Honorius, I recommend changing “St. Honorius” to either “St. Honoratus” or “St. Honoré” throughout the text. 

Plural “women” should be changed to singular “woman” toward the bottom of the paragraph “St. Honorius’ Vita” on p. 3. 

On p. 4, clarify that the trumeau figure of the Vierge Dorée in situ is a copy; the original is inside the cathedral.

“Fouilly” should be “Fouilloy” and “Luzaches” should be “Luzarches” on p. 4, note 16

“Freeze” should be “frieze” and “Richards'” should be “Richard’s” on p. 12

Comments on the Quality of English Language

There are a few problems with prepositions and articles that will undoubtedly be sorted out in copy editing, including: 

"mitred statue of the St." should be "mitred statue of St."(5)

"approaches to the altar" should be "approaches the altar" (6)

"a boy grasped on his arm by a woman" awk (7)

"looking up the tympanum" should be "looking up at the tympanum" (11)

"seminal" word choice (11)

"vita of the St." should be "vita of St." (13)

"entered to the cathedral" should be "entered the cathedral" (13)

Author Response

I wish to thank you for the insightful and useful comments on my article “The St. Honoré Portal at Amiens Cathedral and Its Reception”, which have helped me to further develop and present my arguments in greater depth. I have read all the comments carefully and introduced the changes/additions where needed.

  1. All the missing bibliography and recent studies by Jung, Murray, and Hansen have been added and addressed in the text (pp. 2-3) as well as in footnotes 9, 10, and 12.
  2. I agree that there are more examples of the engagement of the viewer in the western frontispiece. However, the present case study is limited to the south transept portal only, as the study focuses on a specific portal, as suggested in the title, and not on the reception of the entire cathedral. Moreover, the south portal was used by specific audiences that are echoed in both image and text. As such it offers an hermetic case study that functions differently from that of the west façade. I believe that a discussion on the figures of the trumeau or those of the St. Firmin portal (which portrays many inviting figures) would divert the readers’ attention to a different direction than the one suggested in the title. I do, however, believe that a study of the entire reception of Amiens Cathedral is needed, and I feel motivated to further investigate in this direction.
  3. All the specific comments have been dealt with and the article has been proofread by an English language editor.

Round 2

Reviewer 1 Report

Comments and Suggestions for Authors

 This is a well-researched essay with a sophisticated and nuanced argument. It deserves publication.

I found one typo that has to be deleted. Please delete the word "all" in the following sentence. (p. 14)

Moreover, the fact that the child is not depicted as crippled implies both that the miraculous cure was successful and that all every child in need can participate in the processions and give reverence to the saint. 

Author Response

Dear reviewers,

I wish to thank you again for the time and efforts while reading my paper “The St. Honoré Portal at Amiens Cathedral and Its Reception.” The comments were useful and helped me developed and deepen my arguments and thus writing more thoroughly. I read them carefully and integrated the necessary changes where needed.

In page 14: I deleted the unnecessary word “all” in the phrase.

Again, thank you for accepting this paper.

All the best,

Author